# Using Product/Service-System Family Design for Efficient Customization with Lean Principles: Model, Method, and Tool

**Tomohiko Sakao** [1,*], **Tatsunori Hara** [2] **and Ryo Fukushima** [2]

[1] Division of Environmental Technology Management, Department of Management and Engineering, Linköping University, SE-581 83 Linköping, Sweden

[2] Graduate School of Engineering, The University of Tokyo, 7-3-1 Hongo, Bunkyo-ku, Tokyo 113-8656, Japan; hara@tqm.t.u-tokyo.ac.jp (T.H.); fukushima@race.t.u-tokyo.ac.jp (R.F.)

[*] Correspondence: tomohiko.sakao@liu.se; Tel.: +46-73-620-9472

**Abstract:** Facing the need to address environmental issues of our society and individual customer needs and wants along with the trend of offering hybrids of services and products, the ability to efficiently design hybrid offerings is imperative to provide high levels of added value. This ability is much needed throughout the industry, while this type of design is highly challenging due to its high complexity. To date, however, there have been only a few proposals for methods that tackle this challenge at the conceptual design stage. This article proposes a model, a method, and a computerized tool that together support the conceptual design of families of product/service systems (PSSs). First, a PSS family model is presented and then implemented as an add-on to an existing computer-aided design (CAD) tool. Next, a method building upon the model and lean principles is developed as a design procedure. The software and the method are verified through an industrial example of designing a family of logistic services. The proposed model, method, and tool were found effective for describing different key elements of PSS family design. The power of the CAD tool was also demonstrated by taking advantage of a database of model building blocks and semi-automatic calculations.

**Keywords:** customization; integrated solution; CAD tool; architecture; platform

---

## 1. Introduction

To maintain competitiveness in the global market, many companies strive to provide high added value. To add value, it is often imperative for a firm to address individual customer needs and wants, and then design and deliver customized offerings by introducing variety in design [1]. This trend can be widely seen in industry and has been gaining more attention partly owing to increased connectivity with products and end users [2]. But, increasing variety does not always lead to enhanced economic performance [3]: increased variety in products and processes leads to increased complexity and then may end up with decreased competitiveness [4]. To avoid this situation, it is effective to holistically customize a whole group of various products: product family design [5,6] is a major means to this end. Concerning environmental sustainability, the customization of products may or may not entail increased environmental loads and, hence, is an interesting issue.

In relation to customization, industries face a trend of servitization [7]: in today's manufacturing industry, numerous companies' business offerings have become an integration of physical products and services. Here, "service" includes operation, maintenance, repair, upgrade, take-back, refurbishing, remanufacturing, and consultation. Providing services together with products can be an efficient means for the customization of products [8]. This trend has been attracting the attention of a wide

range of industries [9] and academia for the last two decades [10]. These two characteristics of offerings—customized and including service—lead companies to be interested in the customization of PSSs [11,12].

PSS family design is a highly challenging task in design and management because of its high complexity; its design space is enlarged as a design object containing both products and services aiming for multiple customer segments. Further, changes in customer needs over time need to be managed. Namely, many dimensions need to be incorporated for improving the entire PSS family, which is not systematically implemented in industry according to literature [5,12,13]. This high complexity makes computer support for PSS family design effective and promising. Regarding computer support for customizing PSSs, to date, a few models and methods have been reported in the literature [14]. However, virtually no computer tools for operationalizing these models or methods are available. This prevents a firm from even effectively and efficiently customizing PSSs, e.g., with a short lead time, which hinders the realization of high added value. This problem, considering the two industrial trends mentioned above, is expected to become even more conspicuous in the future.

Therefore, this study set out to propose a formal model for PSS families, a practical method for PSS family design, and a computer support tool that semi-automates the entire method by incorporating input from designers on a company's strategies. Instead of PSS customization, PSS family design will be targeted since it aims to improve the entire set of offerings holistically. Also, the method will build upon lean principles [15,16]. The method and tool would be academically innovative as well as practically useful by enabling companies to design offerings with a more holistic view and more efficiently. To do so, the design object model and, in part, the method were implemented as an add-on to existing CAD software for PSS design. The entire software package was verified by being applied to an industrial case of customizing PSSs in the logistics sector.

The remainder of this article is laid out as follows. Section 2 explains the related work and research motivation, followed by the research approach and procedure described in Section 3. Then, Section 4 proposes a formal and comprehensive model and a method for PSS family design, and also describes the implementation. Section 5 describes the application of the model, the method, and the implemented tool to the industrial case. Finally, Section 6 discusses the results, and Section 7 concludes the paper.

## 2. Related Work and Research Motivation

PSSs have been heralded as one of the most effective instruments for moving society towards a resource-efficient, circular economy [17]. Service activity is increasingly incorporated into the design space, which has traditionally been the domain of physical products in the manufacturing industries. In response to the trend toward offering services in the manufacturing industry, several models [18,19] and methods [20,21] for the design and management of PSSs have been proposed; see reviews of PSS design [8,22,23]. Only a few examples of software for the design of PSSs were found, even at the research level: a service CAD integrated with a lifecycle simulator [24], a service CAD called "Service Explorer" [25,26], a support system named PSS-CAD [27], and a modelling tool for PSS Engineering [19]. Implemented and used effectively in practice, these tools will be powerful for enhanced transparency and sharing information across different departments of a PSS provider, and thereby contribute to solve a major challenge of PSS design, that is, making decisions not in silos but globally [28].

For customization of either physical products or services, different models, methods, and tools have been reported in the extant literature [12]. For product customization, a generic customization process based on customization problem-solving situations has been proposed [29]. Service customization has also been tackled, although it is not yet as mature as product customization. For example, software-based service customization models have been developed [30]. Concerning environmental sustainability, the question of the environmental consequences of customization has been asked [31]. Positive factors such as the reduction of unrequired components can be raised [32], while a higher variety of components could lead to more resource consumption [33]. Analysis of customization

in the fashion sector reported decisional areas that included supplier selection and manufacturing defects [34]. The impact of customization on environmental impact, however, is still an open question.

A major key to successful customization is the degree of customization in the entire group of design objects. The concept of mass customization (MC) [35,36], which is already prevalent throughout industry, emerged to address this challenge by being positioned between the mass production and individualized production. For instance, an information system for enabling agile interaction between companies and customers for MC was suggested [37]. More directly, product family design was researched [5,6] aiming to answer the question on customization degree: For instance, a model named the "product family architecture" and its associated methods were developed [38]. A method to solve the product family design problem with selecting a product platform has been developed [39]. These elegant methods are highly useful for optimization when the entire set of products at a point in time is informed to, and under control of, a decision maker. However, companies in reality suffer from product proliferation partly resulting from more ad-hoc customization [40–42]. Several hands-on strategies to reduce product proliferation have been recommended, including use of a cross-functional team to prune off unimportant variants and implementation of effective product withdrawal strategies [40]. Here, PSS providers have an additional opportunity to reduce the proliferation by exchanging products with services. This implies advantages of PSS family design, for which a systematic method is not found in the literature [5,12,13].

Lean principles [15,16] are highly related to family design in a number of ways [43,44]. Lean principles have also been analyzed in terms of their environmental implications in the literature. The lean and green paradigms overlap each other in, e.g., waste reduction [45]. Lean operations could result in sustainability outcomes, including environmental ones [46]. But, not all lean interventions result in environmental gains, as reported in tactical supply chain planning [47]. These variant outcomes are natural since their goals and scopes are different [48] (e.g., lean does not necessarily require the product lifecycle perspective). However, lean principles have the potential to guide family design towards decreased environmental loads by eliminating, e.g., *muda*, that is, what is unnecessary. Further, considering industrial implementation of the sustainability aspect, lean has pragmatic advantages. It is widespread in industry and embedded in its many core businesses. The green concept may come as a natural extension of lean in industrial practice [45], and synergetic effects between the two are expected [49].

Lean principles have been applied to PSS research as well. For instance, Mourtzis et al. [50] proposed a methodology for producing leaner PSS offerings on both process and factory levels, by combining real-time monitoring and correlation analysis with lean principles. Elnadi et al. [51] proposed how to assess leanness of manufacturers' process to provide PSSs without a concrete support for designing PSSs. Resta et al. [52] proposed a theoretical framework for lean operations focusing on product-oriented PSSs: one of the 12 characteristics in the framework concerns product/service range and points out the importance of standardized yet customizable products with a variety of choices of supporting services (also aimed for standardization). Mourtzis et al. [53] proposed a framework for sentiment analysis on the data from manufacturing processes and customer feedbacks together with a set of lean rules for PSS including design aspects that aims to reduce wastes, among others, waiting; e.g., use standardized components. Pezzotta et al. [21] proposed a PSS engineering methodology linked with lean rules on a highly abstract level. The last three papers at least partially addressed design aspects but are limited in providing a concrete support for conceptual PSS design where a higher degree of freedom is available and *exchangeability* between products and services [54] can be used.

Based on the literature briefly reviewed above, this research tackles PSS family design, which is an open research topic [12] and aims to fill the lack of a practical method and tool. Because of its openness to the topic, high-level principles are adopted and borrowed from lean to realize far-reaching effects. In addition to PSSs' potential for positive impacts on the environment, the commonality of lean and green is expected to guide customization to arrive at decreased environmental impacts.

## 3. Research Method

This research builds upon different categories of existing research in terms of the design object and design activity, as depicted in Figure 1, in light of their usefulness and feasibility. The use of an existing CAD tool for PSSs is nominated based on its potential and is adopted for this research (indicated by (a) in Figure 1). Rather than developing the model and tool from scratch, the authors instead took advantage of existing insights as much as possible. Literature related to customization, which covers both product and service customization, was utilized (see (b) in Figure 1).

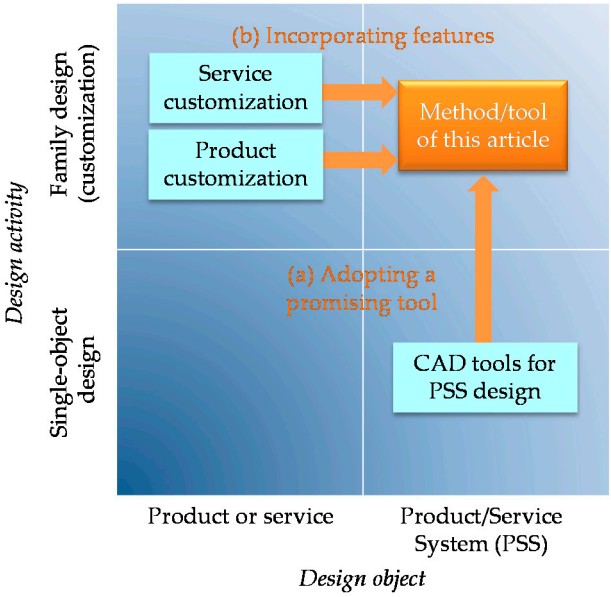

**Figure 1.** Research approach.

Service Explorer was found suitable as a base for the development because of its features for modeling and earlier thorough validation: The latest architecture [25] was built using the eclipse modeling framework (EMF) and graphical modeling framework (GMF), which help to define a structured meta-model for each domain and generate source codes with high operability from the meta-model in the Java programming language. Service Explorer has a hierarchical modeling scheme of functions ranging from means functions that meet with customer requirements and end functions embodied by products and human elements. Parameter constraints have also been introduced to Service Explorer so that the software can detect and resolve conflicts during the process of designing integrated solutions [55]. This article adapts the modeling method implemented for Service Explorer as a basis for the customization of PSS. The details are explained in Section 4.

This research approach was then transformed into a research procedure consisting of three phases, shown in Figure 2. First, in the model and method development, a comprehensive model is synthesized, incorporating the model for designing PSSs from the literature as a base and the useful features extracted from the insights for customization from the literature. Second, in the implementation, a comprehensive model for customizing PSSs is implemented in software building upon the existing Service Explorer. Third, in the verification, the method and the model, implemented in software, are verified in terms of the soundness of their logic and data structure. In addition, an industrial example, based on public data sources from, e.g., companies' brochures, websites, and scientific publications, is examined to verify the method and the model and to illustrate its relevance to practical applications.

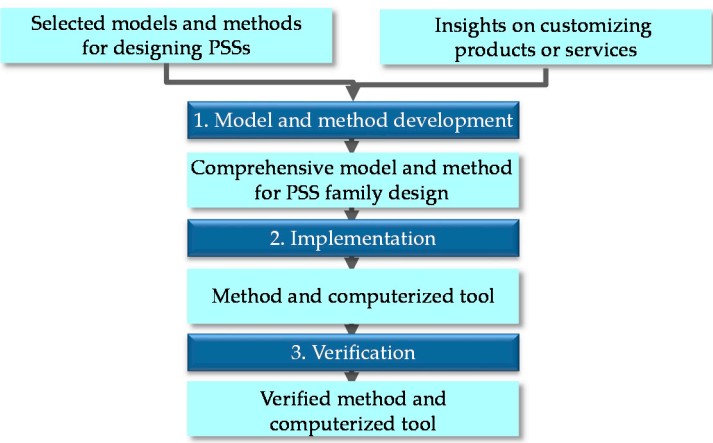

**Figure 2.** Research procedure.

## 4. Model, Method, and Tool for PSS Family Design

### 4.1. Object Model for PSS Family

Figure 3a represents a relevant part of the meta-model in the previous modeling method [25], in which all *functions* and *entities* are defined and instantiated for a specific *PSS*. A PSS also defines an intended customer as a *persona*. The concept of a "persona" [56] is useful for specifying a receiver's requirements. The requirements of each persona are modeled as *receiver state parameters* (RSPs) to be targeted when providing a service. After defining RSPs, the content of PSS is represented as a set of functions and entities in a hierarchical structure. A designer should evaluate the different functions and attempt to find a functional structure that satisfies the persona's expectations of the related RSP in the best way possible. Functions have attributes of *degree of importance* and are realized by entities through the embodiment relationship between *function parameters* and *attribute parameters*.

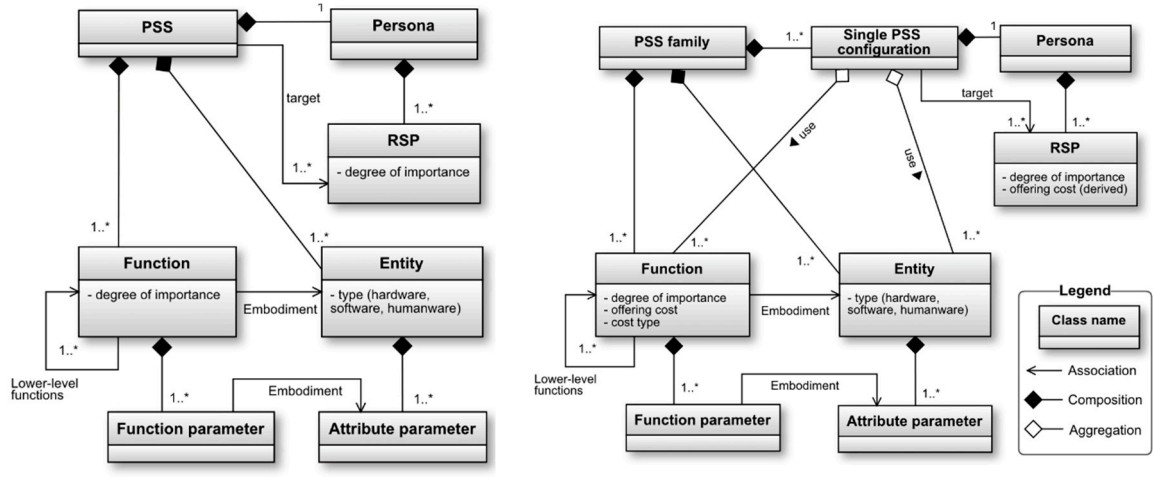

(**a**) Previous study for designing a single PSS    (**b**) This study for PSS family design

**Figure 3.** Proposed meta-model for product/service system (PSS) family compared with PSS.

This paper proposes a new meta-model for PSS family design by incorporating PSS family and configuration features into the previous method, as depicted by Figure 3b. In the new meta-model, the relationship between the different objects can be described as follows: The *PSS family* model holds information about all the possible service and product offerings the provider could offer. A PSS family defines and consists of the *functions* and *entities* available in PSSs currently being designed. Information on each PSS is modeled as a *single PSS configuration* that refers to and uses the available

functions and entities in the PSS family. This supports efficient PSS family design. Two more attributes of functions are explicitly added to the model to evaluate the PSS family: *offering cost* and *cost type*. The details of these are explained in Section 4.2.

Every PSS is ultimately connected to an extended service blueprint [25], consisting of both service activities and product behaviors, outlining how the promised PSS is to be delivered to the customer. The relationship between the functions, entities, service activities, and product behaviors is developed based on the earlier model [25], and the results are used for calculating the cost of each element.

### 4.2. Evaluation of a PSS Family

#### 4.2.1. Overview

A function in a single PSS configuration is likely to affect not only one, but multiple RSPs. In the context of family design, such a function can also be shared and used in different configurations for different customers. This structure of the PSS family is complex for a designer to comprehend, and thus will be a central part to be addressed in the model and the computer tool.

The evaluation method proposed in this section quantifies functions and RSPs with two indices: cost and importance within a PSS family. The evaluation is expected to provide decision support for the designer when developing the PSS family. Crucial here is to address RSPs and functions in evaluation instead of physical products and services because evaluation in this abstract level gives designers the freedom to adopt suitable, concrete means, be it product or service, and this is a major advantage of PSS design termed *exchangeability* [54].

Table 1 illustrates the delivery of a PSS family to *m* customer segments $P = \{p_m \mid m = 1, 2, 3, \cdots\}$, and *n* RSPs $R = \{r_n \mid n = 1, 2, 3, \cdots\}$. The definition of each evaluation is explained below. Note that all of the functions *F* listed here consists of both abstract functions and embodied functions associating entities, that is, the lowest-level functions in the hierarchical structure of PSS family. As described in Table 1, $F^{p_m} \in F$ represents a set of such functions in a single PSS configuration that affects a certain persona $p_m$. A partial set of $F^{p_m}$ that affects a certain RSP is represented as $F_{r_n}^{p_m} \in F^{p_m}$.

**Table 1.** Representation of function sets in a PSS family.

| | | | A Set of RSPs: *R* | | |
|---|---|---|---|---|---|
| | | $\forall r_n \in R$ | $r_1$ | $r_2$ | $r_3$ |
| A set of personas: *P* | $\forall p_m \in P$ | All of functions in PSS family: *F* | $F_{r_1}$ | $F_{r_2}$ | $F_{r_3}$ |
| | $p_1$ | Functions in single PSS configuration: $F^{p_1}$ | $F_{r_1}^{p_1}$ | $F_{r_2}^{p_1}$ | $F_{r_3}^{p_1}$ |
| | $p_2$ | Functions in single PSS configuration: $F^{p_2}$ | $F_{r_1}^{p_2}$ | $F_{r_2}^{p_2}$ | $F_{r_3}^{p_2}$ |
| | $p_3$ | Functions in single PSS configuration: $F^{p_3}$ | $F_{r_1}^{p_3}$ | $F_{r_2}^{p_3}$ | $F_{r_3}^{p_3}$ |

#### 4.2.2. Importance of Function in a Single PSS Configuration

This index is essential for decision making in design in general. The calculation of the relative weight of a function consists of two processes: the significance of RSPs $w_{r_n}^{p_m}$ is analyzed based on customer data and the relevant mathematical techniques (e.g., the analytical hierarchy process (AHP) [57]), and the obtained RSP significance is converted to the function's weight according to the quality function deployment (QFD) method (as implemented on Service Explorer) [58]. Here, let the relative weight of the function $f_k$ on RSPs in a specific PSS be $w_{f_k}^{p_m}$, which is obtained by the method above.

#### 4.2.3. Cost of Function in PSS Family

Since a PSS consists of both physical and nonphysical entities, the costs of different functions can vary in several ways. The proposed evaluation is based on different archetypes for cost variations

in the early stages of the design. Here, three different types of cost are used, as shown in Figure 4: proportionally variable, fixed, and semi-fixed according to a marginal increase of the entity to the associated function. In other words, simplified means of calculating costs were chosen for this study, even though there are other means with higher preciseness available, such as activity-based costing (ABC), which is also implemented in Service Explorer [59]. The designer assigns cost types to functions and defines how the cost varies when functions are added to single PSS configurations. The total offering cost of function $f_k$ within the entire PSS family is represented as $oc_{f_k}$.

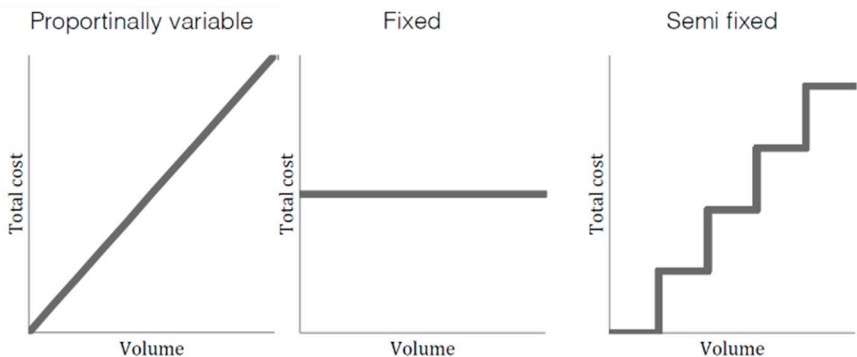

**Figure 4.** The three cost types.

### 4.3. Method for PSS Family Design

#### 4.3.1. Overview

The method presented in this article aims to increase the total customer value over the provider's cost of a given PSS family. To do so, the designer primarily attempts to decrease the cost. This means that increasing the value is out of the scope—this is also why this method is based on the lean approach (to be depicted by Figure 5). In addition, a PSS family involves existing components of products and services of the provider, and implementing a completely new PSS family is unrealistic for convincing customers and too risky for a provider. Therefore, incremental redesign based on an existing solution is applied as an approach. Yet, the operation (shown by Figure 6) maintains the overall aim and minimizes the risk of sub-optimization. Major inputs to the operation are the object model for a given PSS family, as explained in Section 4.1, as well as the importance of functions and the provider's quantitative costs, as explained in Section 4.2.

The first priority is put on removing *muda*, i.e., waste, because waste should always be removed to decrease the cost. *Muda* is especially relevant in the context of PSS family design because recognizing different needs of multiple customer segments may lead to uncovering products or services provided as a standard to but not needed by partial segments. In this type of case, those products or services could be removed from the segments, which leads to customization (see Case 1 in Figure 5). The second priority is on removing *mura*, meaning unevenness (note that some literature has typos, such as mora [60], which does not exist in the Japanese language). Removing *mura* increases the commonality of products or services and a dominant strategy in this operation. Even in the context of customization, adopting common products or services across multiple segments is effective in decreasing the cost via economies of scale. Furthermore, selecting cheaper products or services as common ones will contribute to further reduction of cost. In the case that the products or services are provided to every segment, they are implemented as a platform (see Case 2 in Figure 5). A possible remaining problem to be solved for a PSS family where *muda* and *mura* do not exist is the risk of *muri*, i.e., unreasonableness, in terms of cost. In the Toyota Production System, *muri* originally refers to a situation where too many assignments (i.e., overburden) are given to a shop floor employee, while there is a simultaneous recommendation to avoid hiring an additional employee for the excess bit of the job that is insufficient for one employee, thus unreasonably increasing the cost with the cost type "semi-fixed (S)" [61].

The problem here is the bit in the PSS family with the cost type "semi-fixed" may cost an additional step, and it could be solved by implementing the bit by other means with the fixed or proportionally variable cost types. This could replace a cost-adding product/service with a non- or less-cost-adding product/service. Furthermore, selecting cheaper means, among other things, will contribute to a further reduction of cost (see Case 3 in Figure 5). Figure 7 depicts the state before and after an operation of this sort.

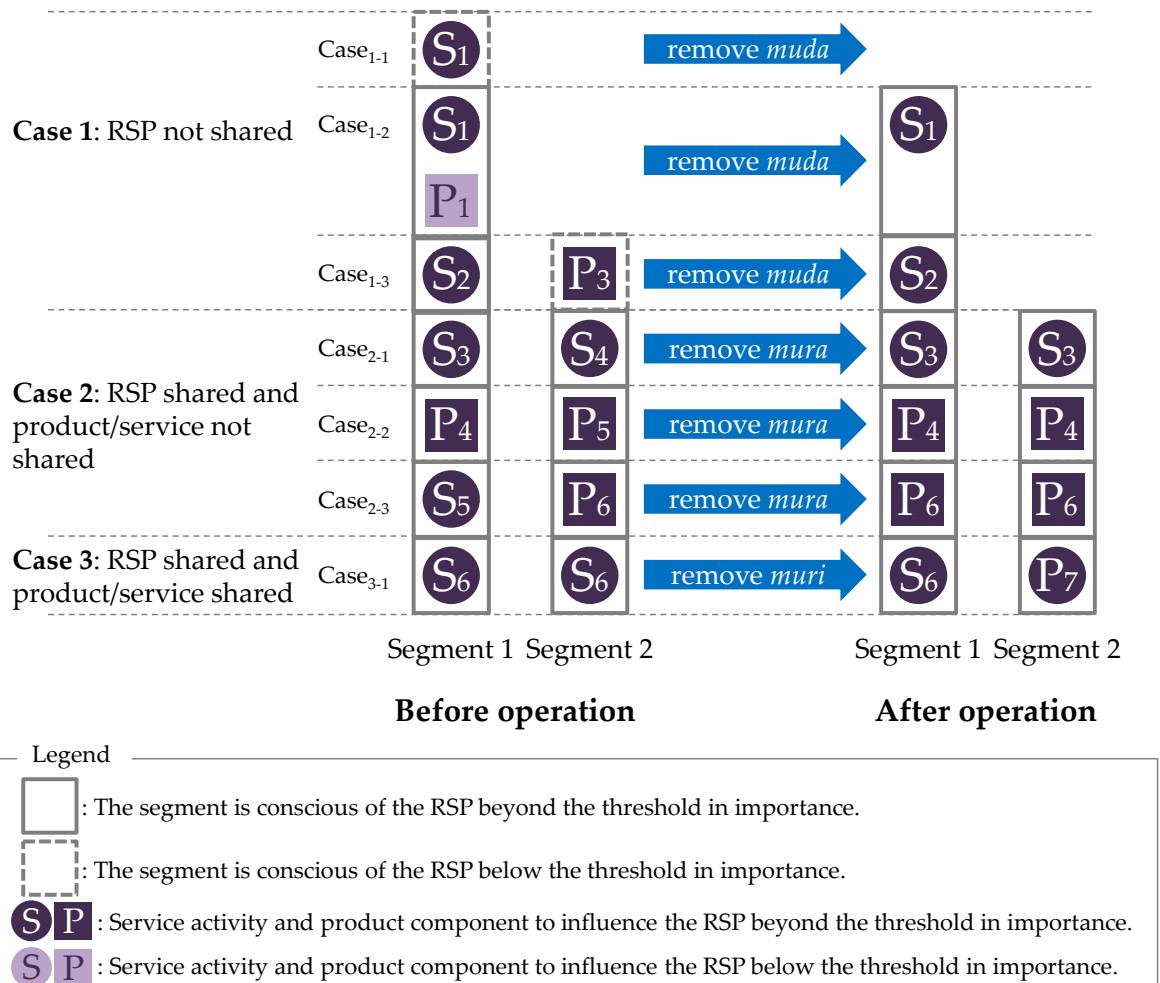

**Figure 5.** Products/services before and after removing *muda*, *mura*, and *muri* within PSS family.

### 4.3.2. Procedure Explained

The proposed method is described as a procedure and is depicted in Figure 6 and described in detail here. This procedure is to be carried out when changes in customers, customers' RSPs, realized products/services, or costs for the products/services are observed. First, the latest customer value is sensed and identified, corresponding to Step 0. This is important because new customers may be added, some customers may not exist anymore, and customers' needs and wants evolve over time. After Step 0, the operation is implemented according to the priorities to different degrees, as described in Section 4.3.1. The reason why the procedure consists of three steps (after Step 0) in a sequential manner is to avoid a massive increase in the number of possible alternatives due to possible combinations of removing *muda*, *mura*, and *muri* in different orders. Each of the steps and sub-steps is described below.

**Step 0. Identify current customer value**—current customer segments are identified, and important RSPs of each segment are selected. This selection can be carried out with a given threshold on the importance of each RSP, but a standard description for this selection is outside the scope of this paper.

**Step 1. Remove *muda* (waste)**—products or services that are implemented for a segment but do not influence any RSP of the segment are removed, as depicted by Case 1 in Figure 5. For instance, in Case$_{1-1}$, Segment 1 is not sufficiently conscious of the RSP, and therefore Service 1 influencing this RSP is regarded as waste. This removal omits the cost of the products or services and decreases the total cost. The decision is made by the designer considering the importance of the RSP and the function in question. *Muda* is addressed as the first step because of the higher priority than *mura* and *muri*, as mentioned above.

**Step 2. Remove *mura* (unevenness)**—this is applied for an RSP, which multiple segments are conscious of, aiming to adopt identical products and services as frequently as possible, as depicted by Case 2 in Figure 5. Step 2 is broken down into the following four sub-steps:

**Step 2-1. Select an RSP in segments**—this picks up one of the RSPs which exist in multiple segments. If there is any difference in products/services that influence the RSP, Step 2-2 is taken. For instance, in Case 2–3, the RSP which Segments 1 and 2 are conscious of is selected, and the difference in the structure (Service 5 and Product 6 for Segments 1 and 2, respectively) is observed.

**Step 2-2. Detect *mura* elements**—this identifies products or services that can be replaced so that identical structure is obtained for the RSP in multiple customer segments. For instance, in Case 2–3, Service 5 or Product 6 is detected for potential replacement, as depicted in Figure 5.

**Step 2-3. Calculate costs**—this calculates the cost for each set of resulting products/services from Step 2-2. For instance, in Case 2–3, the costs of the two options are calculated, where Service 5 is replaced with Product 6 and vice versa.

**Step 2-4. Examine options**—this aims to select an option to be executed by examining possible effects to the RSP and cost calculated in Step 2-3 for each option. This selection is conducted by the designer while considering the balance between the RSP and the decreased cost.

The four sub-steps will be repeated for all the RSPs that exist in multiple segments.

**Step 2-5. Implement the identical structures**—for each of the RSPs selected in Step 2-1, this step replaces the current set of products and services with the identical structure (products and services) by replacing the elements selected in Step 2-4. For instance, in Case 2–3, Service 5 is replaced with Product 6.

**Step 3. Remove *muri* (unreasonable extra cost)**—this is applied for a product or service whose cost is categorized into the type S (semi-fixed), aiming to remove an unreasonable extra cost. Step 3 is broken down into the following five sub-steps. *Muri* is addressed as the last step because of the lowest priority.

**Step 3-1. Select S-type product/service**—this picks up a product or service with the type S (semi-fixed). If there is another way of realizing this by a different product or service with the type F (fixed) or P (proportionally variable), Step 3-2 is taken.

**Step 3-2. Replace the S-type product/service with an F- or P-type product/service**—this identifies the products or services with the type F or P that can be a replacement.

**Step 3-3. Calculate costs**—this calculates the cost for each one identified in Step 3-2. To do so, data on how much the concerned products or services will be increased or decreased is needed and used. Figure 7 shows costs before and after replacement on different cost curves.

**Step 3-4. Select the cheapest**—this selects the least expensive option among the alternatives calculated in Step 3-3.

The four sub-steps will be repeated for all the products and services with the cost type S.

**Step 3-5. Implement the cheapest**—this step replaces each of the S-type products and services selected in Step 3-1 with the cheapest one obtained in Step 3-4, if available.

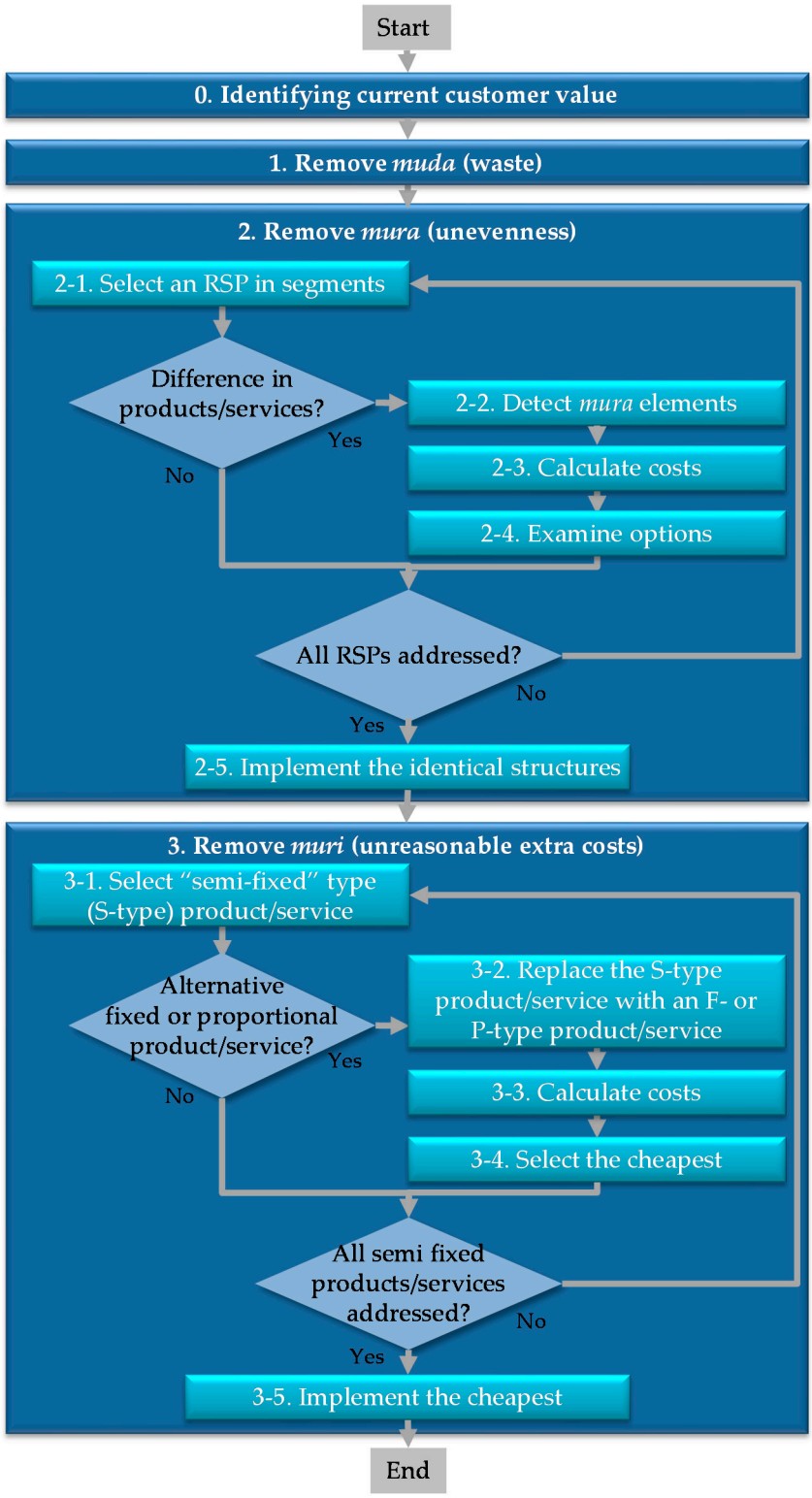

**Figure 6.** Flow chart of the procedure applied to PSS family.

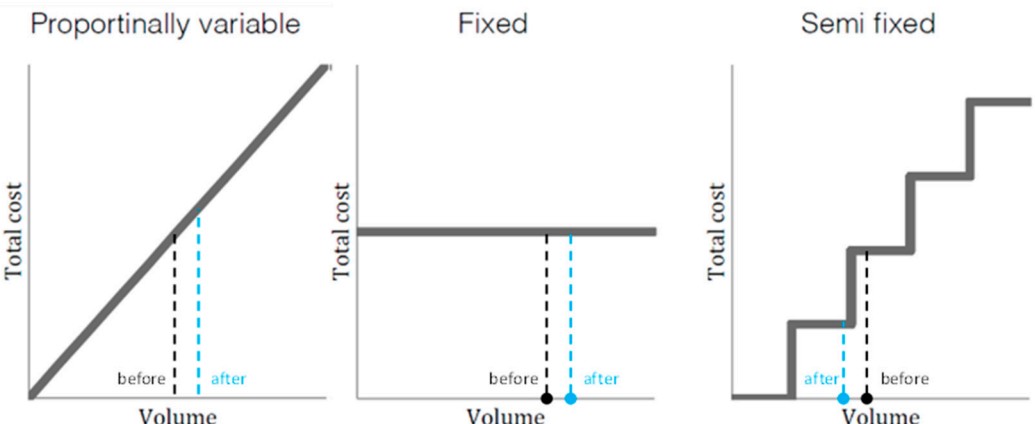

**Figure 7.** Costs before and after removing *muri* from one with the S (semi-fixed) type cost.

### 4.4. Implementation as Software

#### 4.4.1. Overview

According to the proposed model in Section 4.1, as well as the method in Sections 4.2 and 4.3, new implementation on Service Explorer was achieved to support PSS family design. Figure 8 is a screenshot of the newly developed Service Explorer. The upper parts of the figure show the hierarchical function structure for each RSP in a single PSS configuration (i.e., $F_{r_i}^{p_m}$). The developed software visually shows and suggests functions that could be recognized as *muda*, *mura,* and *muri*. These functions are indicated by a question mark in the table (in the importance column) in the lower part of Figure 8. With this interface, designers are supported to make relevant decisions to delete or replace functions recognized as *muda*, *mura,* and *muri* on a given PSS family rather than on a single PSS. Information provided to a designer in relation to *muda*, *mura,* and *muri* is explained in the following sections.

Note that Service Explorer manages identification of, for instance, a function used for different RSPs or for different personas in a PSS family. Although this has been the case in both the previous and the new versions of Service Explorer, this is noteworthy—this is an advantage that is hard to provide with a tool without the formal model.

#### 4.4.2. Removing *Muda*

The functions that have the lowest importance and are judged as possible *muda* are indicated by the software. Designers thus can find functions as possible options to be deleted. While choosing functions to be deleted, cost information shown in the software helps designers evaluate the degree of cost reduction after deleting these functions.

#### 4.4.3. Removing *Mura*

The functions that fulfil the following constraints are both judged possibly as *mura* and also indicated by the software so that designers understand why these are judged as *mura*. This can be achieved by searching throughout a given PSS family that consists of single PSS configurations.

- Provided in a single PSS configuration for a customer segment with persona $p_m$ and an RSP $r_i$ ($f_j \in F_{r_i}^{p_m}$);
- Have *sibling* functions that realize the same upper function (i.e., the *parent* function) in a single PSS configuration for another customer segment with persona $p_n$.

$$\exists f_{sibling} \in F_{r_i}^{p_n} \left( n \neq m, \; upperof(f_j) = upperof(f_{sibling}) \right)$$

Further, referring to information of cost and importance shown on the software, designers are supported to decide if they remove *mura*.

### 4.4.4. Removing *Muri*

Functions with their cost being of the semi-fixed type are judged to have the possibility of *muri* and shown as such. In addition, designers can see the prospect of the cost reduction on the software by removing *muri* (as shown in the cost graph for a function and tableau named "detail of *muri*"); see Figure 8.

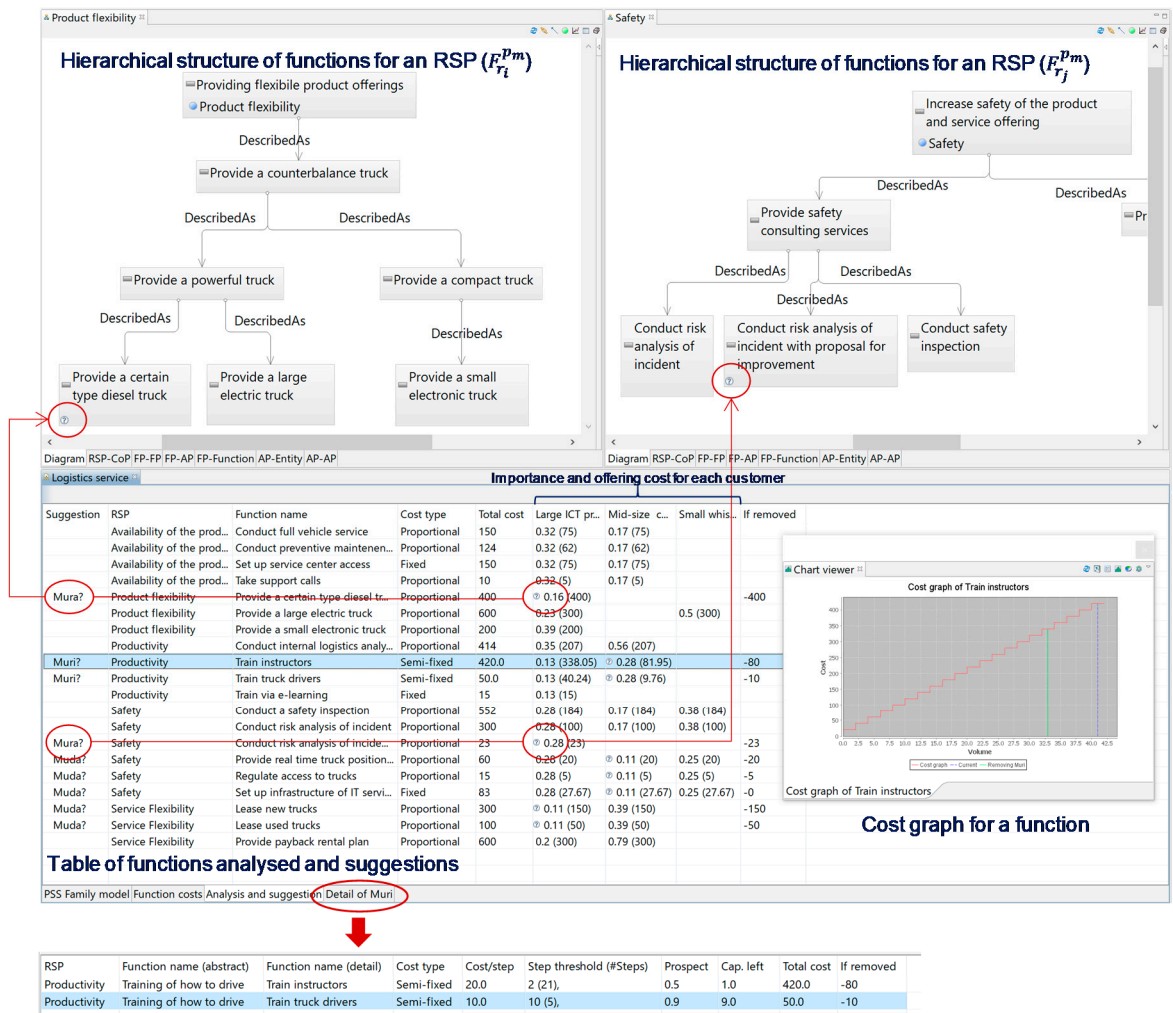

**Figure 8.** Screenshot of function analysis in PSS family design in the software. Notes: In the table of functions (middle), the functions judged possibly as *muda*, *mura*, or *muri* are indicated in the far left-hand-side column while the importance of each function for a relevant customer is annotated with a small question mark (e.g., the function "Provide a certain type diesel truck" was suggested as *mura* in the function structure for Customer I). Among them, the possibility of *mura* is also indicated in the visualized function structure with the same question mark (in the upper parts of the figure, "Hierarchical structure of function for an RSP"). This is implemented so that designers can find siblings of the function that share the same parent function during the process of removing *mura* (e.g., "Provide a large electric truck" is a sibling node of "Provide a certain type diesel truck" suggested as *mura*).

## 5. Application to Industrial Case

*5.1. Case Data*

Section 5 examines the case of a logistics service provider and aims to illustrate an industry-sourced case of PSS family design, where the proposed model, method, and tool are applied. With this aim in mind, the authors collected data for the PSS family from multiple companies operating in this sector. The majority of the data sources was companies' brochures and websites, as well as scientific publications relevant to the sector. When the required data were unavailable, an assumption was made by the authors.

The PSS family provided by the provider addressed in this case consists of a wide range of services, ranging from rental solutions and driver training support to materials handling consultation, as well as materials handling equipment such as warehouse trucks, counterbalanced trucks, and semi-automatic materials handling systems [62]. The revenue ratios of both services and customized offerings in this sector are substantial in Europe. For instance, the revenue ratio of a company's rental offerings is estimated as being 50 to 60% [63,64], and that of customized offerings is approximately 40% [64].

*5.2. Modeling the Current PSS Family*

5.2.1. Personas and RSPs

The personas in this B2B (business to business) case are representations of actual users such as a company or an employee, e.g., one of the truck drivers or the person in charge of warehouse management. Customers were chosen to represent a wide range of different use cases and customer needs, and with that in mind, companies of different scales from different sectors were selected. These customers are presented below with a brief description based on public brochures [65–67] (see also Figures A1–A3 in Appendix A). We associated RSPs with the personas based on each customer's different needs. All of the RSPs selected for the customers are listed in Table 2. The weight of an RSP reflects its importance. The total significance of RSPs for each customer was set in proportion to the hypothetical revenue from the customer, which reflects the degree of benefit delivered to the customer.

**Small whisky producer (Customer W)**—This customer requires trucks to operate safely and smoothly since it carries high-value loads, but it does not need a large number of trucks since its operation is still small-scale. Within this small company, the truck operator will need a small and safe truck that can easily navigate the underground environment in which it stores its products. Product flexibility should be important since only one truck will be deployed by the customer.

**Mid-size construction company (Customer C)**—This is a manufacturer of, among other things, concrete slabs for groundworks. These concrete slabs are heavy and therefore require the use of heavy-duty trucks for the materials handling. It has a large fleet of vehicles and is interested in using the provider's payback scheme, where it sells its trucks to the provider and then rents them back in order to free up some of its assets. Since its production is highly seasonal, this would help it cut costs during the less productive autumn and winter months when it does not have any products in its warehouse. To navigate the warehouse, the customer needs trucks with a slimmer profile than what is usual for counterbalance trucks. The customer also manufactures its own pallets for certain goods, which also places higher demands on the trucks that transport them.

**Large ICT product provider (Customer I)**—This is an ICT (information and communications technology) manufacturer of telecom equipment, which places a high value on security. New drivers should always have a truck-driving license; otherwise, they must be educated on-site. Only authorized drivers should be allowed to operate in the warehouse, and a senior employee often accompanies new recruits. Because safety is key to the operations of the customer, it will require a large number of training sessions, instruction, and consultations to ensure that its warehouse and staff can comply with the regulations that it has in place.

**Table 2.** Selected receiver state parameters (RSPs) for the three case customers.

| Customer | 1. Availability | 2. Safety | 3. Productivity | 4. Product Flexibility | 5. Service Flexibility |
|---|---|---|---|---|---|
| Small whisky producer (Customer W) | | 0.8 | | 0.2 | |
| Medium-sized construction company (Customer C) | 0.5 | 0.6 | 0.8 | | 0.8 |
| Large ICT product provider (Customer I) | 0.8 | 1.0 | 0.8 | 0.6 | 0.4 |

### 5.2.2. Functions and Attributes

A number of functions and attributes in the PSS family was modelled according to the three personas and the five RSPs described above. The personas, RSPs, and functions were limited in numbers, in order to be easier for a reader to grasp this case. These functions are summarized in Table A1 in Appendix A. Most of the functions are provided to multiple customers, providing high potential of family design. The functions labelled with "(abstract)" directly influence an RSP, such as the provision of a truck or consulting services. The lower functions are those that indirectly affect an RSP through content. A lower function describes or decomposes its parent. The weights of functions are obtained from the significance of RSPs according to the QFD method, as explained in Section 4.2.2. The costs for each RSP are hypothetical and do not represent the actual costs for any product or service. The total offering cost of functions in Table A1 is calculated considering the cost type, times used in the PSS family, and marginal cost.

### 5.3. Operations of PSS Family Design

With the modeled PSS family, including the cost information and the importance of functions, the developed software analyzed the model and suggested possible choices of removing *muda*, *mura*, and *muri*; see the first column of Figure 9.

### 5.3.1. Step 1—Remove *Muda*

Five functions were found to have the lowest importance (i.e., 0.1) in all the functions, namely two functions for Customer I, "Lease used trucks" and "Lease new trucks" and three functions for Customer C, "Set up infrastructure of IT services", "Regulate access to trucks", and "Provide real-time truck positioning data" in Figure 9. The designer made a decision concerning these five functions, as described in Table 3. The designer, regarding the cost type, recognized that the function, "Set up infrastructure of IT services" (#3), should not be deleted since the total offering cost would not change by deleting this function with the cost type "fixed" (see Table A1). On the other hand, by deleting the other four functions with the cost type "proportional", the total offering cost would be decreased.

Figure 9 shows the offering cost of these five functions for each customer (next to the importance information) that is assumed identical among customers. In comparing the five functions in terms of their cost in Figure 9, the designer decided to delete "Lease new trucks" (#2), since the offering cost of the function was the highest in the four functions (see Table A1). After this step, the total cost was decreased from 4976 to 4826. The other four functions were decided to be kept because they had less cost and non-negligible importance.

Logistics service ⊠

| Suggestion | RSP | Function name | Cost type | Total cost | Large ICT pr... | Mid-size c... | Small whis... | If removed |
|---|---|---|---|---|---|---|---|---|
|  | Availability of the product and service | Conduct full vehicle service | Proportional | 150 | 0.22 (75) | 0.12 (75) |  |  |
|  | Availability of the product and service | Conduct preventive maintenence | Proportional | 124 | 0.22 (62) | 0.12 (62) |  |  |
|  | Availability of the product and service | Set up service center access | Fixed | 150 | 0.22 (75) | 0.12 (75) |  |  |
|  | Availability of the product and service | Take support calls | Proportional | 10 | 0.22 (5) | 0.12 (5) |  |  |
| Mura? | Product flexibility | Provide a certain type diesel truck | Proportional | 400 | ⑦ 0.12 (400) |  |  | -400 |
|  | Product flexibility | Provide a large electric truck | Proportional | 600 | 0.18 (300) |  | 0.2 (300) |  |
|  | Product flexibility | Provide a small electric truck | Proportional | 200 | 0.3 (200) |  |  |  |
|  | Productivity | Conduct internal logistics analysis | Proportional | 414 | 0.4 (207) | 0.4 (207) |  |  |
| Muri? | Productivity | Train instructors | Semi-fixed | 420.0 | 0.13 (338.05) | ⑦ 0.2 (81.95) |  | -80 |
| Muri? | Productivity | Train truck drivers | Semi-fixed | 50.0 | 0.13 (40.24) | ⑦ 0.2 (9.76) |  | -10 |
|  | Productivity | Train via e-learning | Fixed | 15 | 0.13 (15) |  |  |  |
|  | Safety | Conduct a safety inspection | Proportional | 552 | 0.17 (184) | 0.15 (184) | 0.2 (184) |  |
|  | Safety | Conduct risk analysis of incident | Proportional | 300 | 0.17 (100) | 0.15 (100) | 0.2 (100) |  |
| Mura? | Safety | Conduct risk analysis of incident with proposal ... | Proportional | 23 | ⑦ 0.17 (23) |  |  | -23 |
| Muda? | Safety | Provide real time truck positioning data | Proportional | 60 | 0.17 (20) | ⑦ 0.1 (20) | 0.13 (20) | -20 |
| Muda? | Safety | Regulate access to trucks | Proportional | 15 | 0.17 (5) | ⑦ 0.1 (5) | 0.13 (5) | -5 |
| Muda? | Safety | Set up infrastructure of IT services | Fixed | 83 | 0.17 (27.67) | ⑦ 0.1 (27.67) | 0.13 (27.67) | -0 |
| Muda? | Service Flexibility | Lease new trucks | Proportional | 300 | ⑦ 0.1 (150) | 0.22 (150) |  | -150 |
| Muda? | Service Flexibility | Lease used trucks | Proportional | 100 | ⑦ 0.1 (50) | 0.22 (50) |  | -50 |
|  | Service Flexibility | Provide payback rental plan | Proportional | 600 | 0.2 (300) | 0.45 (300) |  |  |

PSS Family model | Function costs | Analysis and suggestion | Detail of Muri

**Figure 9.** Screenshot of the software suggesting possible choices of removing *muda, mura,* and *muri.*

**Table 3.** Summary of the designer's decisions concerning *muda*.

| # | Function | Customer | Cost Type | Evaluation by Designer | Decision Taken | Note |
|---|----------|----------|-----------|------------------------|----------------|------|
| 1 | Lease used trucks | I | Proportional | Possible option to be deleted | Kept | |
| 2 | Lease new trucks | I | Proportional | Best option to be deleted | Deleted | The highest cost-reduction in the possible options |
| 3 | Set up infrastructure of IT services | C | Fixed | Not to be deleted | Kept | The total offering cost would not change |
| 4 | Regulate access to trucks | C | Proportional | Possible option to be deleted | Kept | |
| 5 | Provide real-time truck positioning data | C | Proportional | Possible option to be deleted | Kept | |

### 5.3.2. Step 2—Remove *Mura*

Two functions, "Provide a certain type diesel truck", belonging to the RSP "Product flexibility", and "Conduct risk analysis of incident with proposal for improvement", belonging to the RSP "Safety", were suggested by the software as *mura*, as shown in the left-hand side column of Figure 9. The designer decided how to implement identical structures for the *mura* functions, as shown below.

Visualized hierarchical structures of functions as shown in the upper parts of Figure 8 supported the designer's decision. For example, Figure 10 shows these structures in single PSS configurations for Customers I and W in the case of "Product flexibility" (i.e., $F_{r_4}^I$ and $F_{r_4}^W$). The designer found the reason for *mura* from the yellow box in Figure 10a, describing that the function "Provide a certain type diesel truck" is used only for Customer I". In order to decide whether an identical structure be implemented or not, the designer was able to check the possible effect to the RSP in Figure 9. Since "Provide a certain type diesel truck" has the least importance (i.e., 0.12 as shown in Table A1) in the RSP, the function could be replaced and unified with the structure of Figure 10b (adopting "Provide a large electric truck.") After this step, the total cost was decreased from 4826 to 4426. The summary of the designer's decision for this operation is described in Table 4.

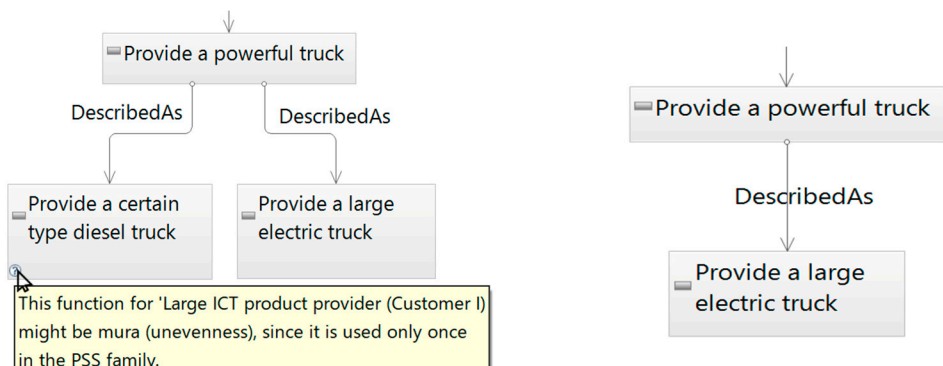

(**a**) Function structure for Customer I ($F_{r_4}^I$)　　　(**b**) Function structure for Customer W ($F_{r_4}^W$)

**Figure 10.** Screenshot of function analysis in PSS family design in the software.

**Table 4.** Summary of the designer's decision concerning *mura* in the case of RSP "Product flexibility".

| # | Function | Evaluation by a Designer | Decision Taken | Note |
|---|----------|--------------------------|----------------|------|
| 1 | Provide a certain type diesel truck | Possible option to be replaced as *mura* | Replaced | Least importance under the same RSP |

Figure 11 shows function structures for Customers I and W in the case of RSP "Safety" (i.e., $F_{r_5}^I$ and $F_{r_5}^W$). Designers found *mura* in the yellow box in Figure 11a, showing that "Conduct risk analysis of incident with proposal for improvement" was used only for Customer I (see Table A1). In order to decide whether this advanced function should be replaced, the designer was able to check the possible effect on the RSP in Figure 9. The designer found that even the two *sibling* functions (i.e., "conduct risk analysis of incident" and "conduct safety inspection") were able to fulfil the RSP without the advanced function since only one-third of importance would be deleted. As above, the designer judged it to be replaced with the two sibling functions. After this step, the total cost was decreased from 4426 to 4403. The summary of the designer's decision of this unification is described in Table 5.

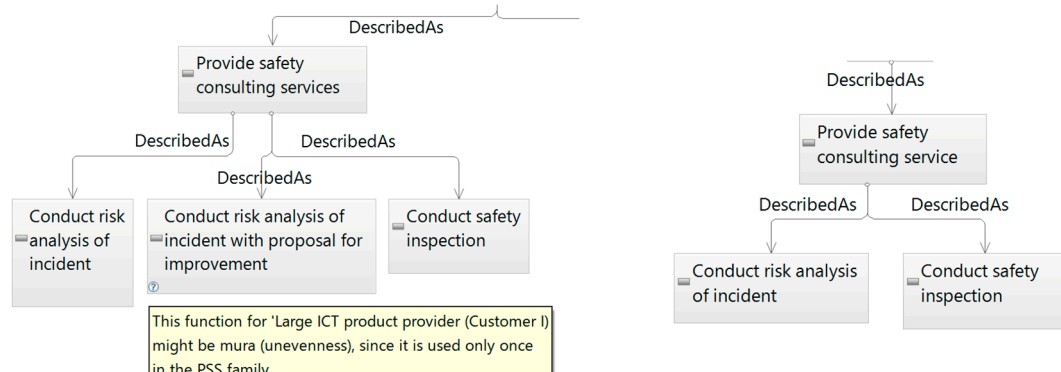

(**a**) Function structure for Customer I ($F_{r_5}^I$)     (**b**) Function structure for Customer W ($F_{r_5}^W$)

**Figure 11.** Screenshots of function structure for RSP "Safety".

**Table 5.** Summary of the designer's decision concerning *mura* in the case of RSP "Safety".

| # | Function | Evaluation by a Designer | Decision Taken | Note |
|---|----------|--------------------------|----------------|------|
| 1 | "Conduct risk analysis of incident with proposal for improvement" | Possible option to be replaced as mura | Replaced | Only one-third of importance could be decreased |

### 5.3.3. Step 3—Remove *Muri*

In the left column of Figure 9, the two functions, "Train truck drivers" and "Train instructors", both of which Customer I and Customer C were provided with, were shown possibly as *muri* since the cost type of these two functions was semi-fixed. The designer checked the details of these possibilities in Figure 12. In this case, the designer learned that "Train truck drivers" would entail higher cost reduction, meaning that the cutback of the function volume (i.e., stop offering the function to Customer C) contributed to cost reduction, as shown in Figure 12. Prospects of cost reductions are calculated as the *capacity left (cap. left)* divided by the *step threshold*.

In order to compensate for the reduced level of the function without increasing cost, the designer found "Training via e-learning" as a compensatory function from Figure 9. Since the function "Training via e-learning" was already offered to Customer I, this function would be used instead of "Train truck

drivers" for Customer C. After this step, the total cost was decreased from 4403 to 4393. The summary of the designer's decision for removing *muri* is described in Tables 6 and 7.

As a result of operations explained in this section, the designer succeeded in an approximate 12% cost reduction in the PSS family (i.e., from 4976 to 4393).

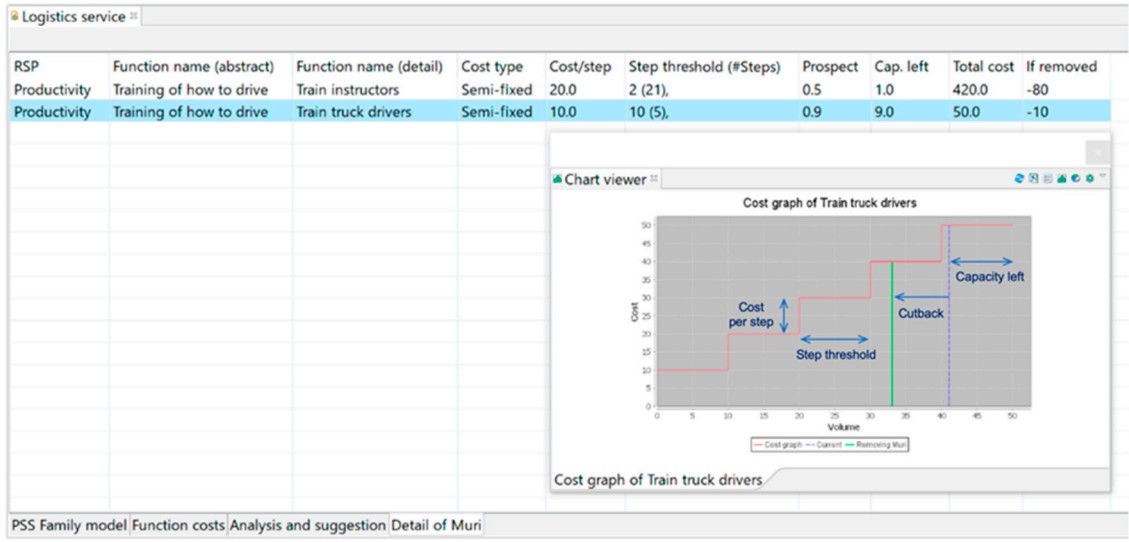

**Figure 12.** Screenshot of inspecting a detected *muri* possibility.

**Table 6.** Summary of the designer's decision concerning cutback of function volume to reduce *muri*.

| # | Function | Evaluation by a Designer | Decision Taken | Note |
|---|---|---|---|---|
| 1 | Train truck drivers | Possible option to be reduced | Reduce function volume for Customer C | Small cutback (volume for Customer C) contribute to large cost reduction |
| 2 | Train instructors | Possible option to be reduced | Nothing | |

**Table 7.** Summary of the designer's decision concerning compensatory function.

| # | Function | Evaluation by a Designer | Decision Taken | Note |
|---|---|---|---|---|
| 1 | Set up service center access | Possible compensatory function | Not chosen | |
| 2 | Training via e-learning | Possible compensatory function | Chosen | Can be utilized as means for driver training |
| 3 | Set up infrastructure of IT services | Possible compensatory function | Not chosen | |

## 6. Discussion

### 6.1. Scientific Contributions

The major scientific novelty of this article exists in the proposal of a model, a method, and a computerized tool for PSS family design. This is supported by a review of the literature in the areas of PSS design and product/service family design (as presented in Section 2). The generic character of this model and thus its broad applicability advances the body of knowledge by building upon the literature in the variety of relevant areas (as explained in Section 3).

The proposed method is original in exploiting the set of *muda*, *mura*, and *muri* from the lean production system [61] in the PSS design context. While managing customization at the right level has

been researched mostly with the platform concept, this article tackles this challenge in such a practical way that the method is considered conceptually easy to be implemented in practice.

By implementing the model in software, the soundness of the logic as well as the data structure of the model were verified. In addition, by using an industrial example (as presented in Section 5), the proposed model and tool were verified with regard to their efficacy. No earlier research in PSS design software is capable of addressing the concurrent design of multiple offerings. Specifically, a designer obtains pointers in the context of the entire PSS family that allows him/her to decide whether they should implement promising design operations for a PSS, such as omitting insignificant functions. Furthermore, the power of the computerized tool is shown by using a database storing various building blocks for such functions.

Noteworthy is the flexibility of the model, which allows designers to describe the functions of either a product or a service and thus to analyze the possibility of substituting a product with a service (and vice versa), as explained in Section 4.3. This enables designers to take advantage of the main feature inherent from the design of a PSS, i.e., the utilization of suitable means, be it products or services (termed *exchangeability* in [54]). This gives additional opportunities to enhance a PSS offering as compared to addressing either products or services separately.

After performing the application (Section 5), a limitation of the method was found. In removing *mura*, the quantitative level of a function in question was found not fully considered. For instance, as shown in Figure 10, "Provide a large electric truck" was adopted for its parent function "Provide a powerful truck" for both Customers I and W after removing *mura* (i.e., omitting "Provide a certain type diesel truck") successfully. The needed quantity of the electric trucks was not considered, according to the method. Therefore, there is a risk that the level of "Provide a large electric truck" provided for Customer I is insufficient after this operation. Developing a method that handles the quantitative levels of functions will be addressed in future work.

Cocreating value according to what is needed or wanted by PSS recipients [20,22], which corresponds to realizing effectiveness, is one of the most important features of PSS design [68]. Value cocreation is a process involving customers and concerning not only customer value but also provider value [69]. It is often more complex due to the additional service element compared to product design. On the other hand, managing efficiency is also important in PSS design by, e.g., addressing the cost [70], as in any engineering activity, and thus has been part of some PSS design methods (e.g., comparison between value and cost in [71]). This article, by filling the gap of PSS family design in the literature, contributes to solve the proliferation problem, aiming to enhance the efficiency with minimizing sacrifice, if not none, in effectiveness. The aim did not necessarily lie in optimisation of configurations but was to provide a more realistic hands-on method in practice. However, PSS family design is now opened up as a relevant research topic and readers are invited to contribute to further debates. One possible approach to avoid excessive compromise in effectiveness is introducing optimisation. Yet, the optimisation in practice is likely to face a challenge to accuracy and consistency of data such as importance of customer needs/wants for each configuration, and in this case incorporating uncertainty could be a mathematical solution.

*6.2. Industrial Relevance*

Section 5 showed that the proposed model, method, and tool are effective when applied to an industrial example when performing PSS family design. The information used for the example is expected to be obtainable from the PSS provider in general and at the conceptual design stage. Thus, the model, method, and tool have wide applicability to PSS family design. In addition, their use in an early stage of design has a major impact on the design efficiency. Further, the tool contributes to a dramatic decrease in the time required to analyze PSS family design, given its automated procedure. In particular, the tool's applicability to the conceptual design stage increases its relevance to industry, especially given the current lack of effective tools for that stage.

*6.3. Environmental Implications*

Positive factors of lean on the environment reported in the literature, such as reduction of *muda* and *mura*, were exploited to propose the method in facing the open question concerning the implications of customization on the environment. Also, PSSs' potential positive contribution to the environment based on, e.g., the lifecycle perspective [17] is embedded in the proposed method. Hence, positive effects with the proposed method on the environment will be expected. Whether or not the actual effects are positive is unknowable at the conceptual design stage, and we need to wait for the outcome of detailed design, production and use.

## 7. Conclusions

This article proposed virtually the first-ever model, practical method, and computerized tool capable of performing PSS family design by employing state-of-the-art PSS design and product/service family design. The proposed model, method and tool were shown to be effective in supporting PSS family design through application to an example set up based on information from industry. The effect of the computerized tool was also shown by automating the calculation procedure and by visualizing the calculation results.

Because of the degree of innovation and the complexity of the challenge and the proposal, this article opens up various avenues for future research besides the development of the method that handles the quantitative levels of functions (stated in Section 6.1). First, implementing other features relevant for customization and family design such as *profit*, *commonality,* and *modularity* [72] into the object model is an immediate task for the future. Second, applying the model, method, and tool to the tasks involved in the day-to-day business of a company is another promising avenue that would further validate the proposal. Third, the environmental implications of PSS customization and family design need more research, including quantitative investigation.

**Author Contributions:** T.S. conceptualized the method with T.H.; T.H. designed the proposed model and computer tool. T.H. implemented the tool with R.F.; R.F. studied the case with T.H.; T.S. led the paper writing. T.S. and T.H. wrote the paper with the help of R.F. All authors have read and agreed to the published version of the manuscript.

**Funding:** This research has received no external funding.

**Conflicts of Interest:** The authors declare no conflict of interest.

## Abbreviations

| | |
|---|---|
| ABC | activity-based costing |
| AHP | analytic hierarchy process |
| B2B | business to business |
| CAD | computer-aided design |
| EMF | eclipse modeling framework |
| GMF | graphical modeling framework |
| ICT | information and communications technology |
| MC | mass customization |
| PSS | product/service system |
| QFD | quality function deployment |
| RSP | receiver state parameter |
| R&D | research and development |

## Appendix A. PSS Family Provided for the Three Customers Referenced in the Case

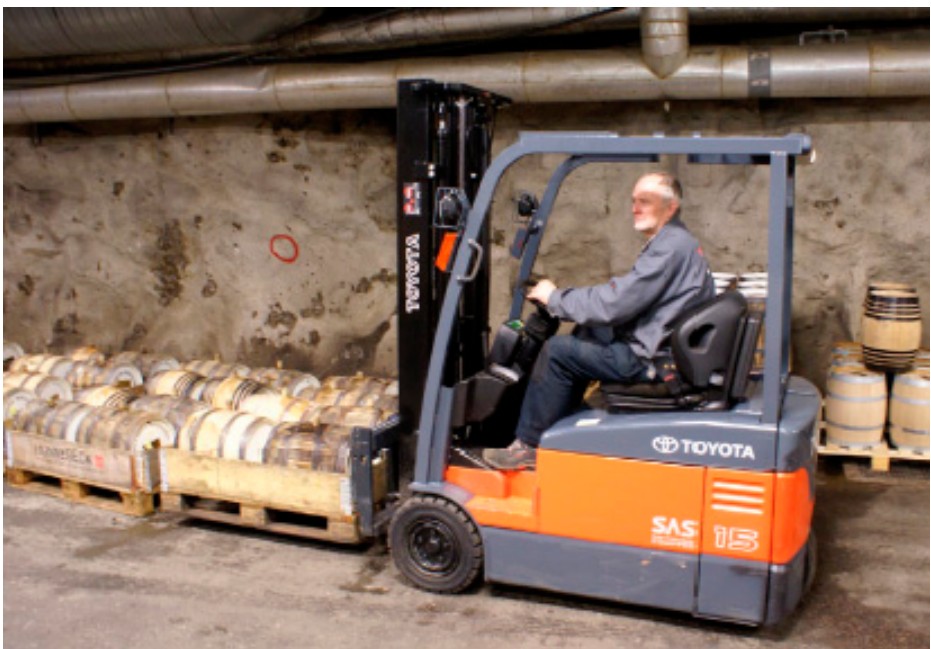

**Figure A1.** A scene of PSS provided for the whisky producer (Customer W) [65].

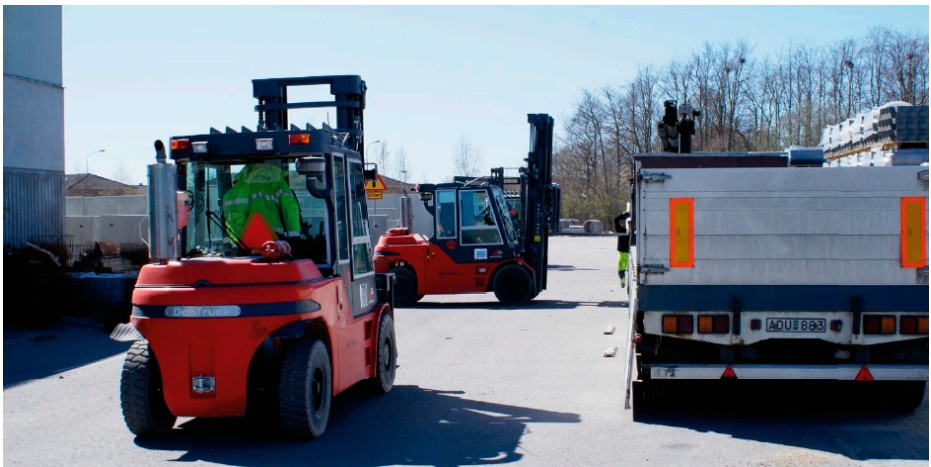

**Figure A2.** A scene of PSS provided for the construction company (Customer C) [66].

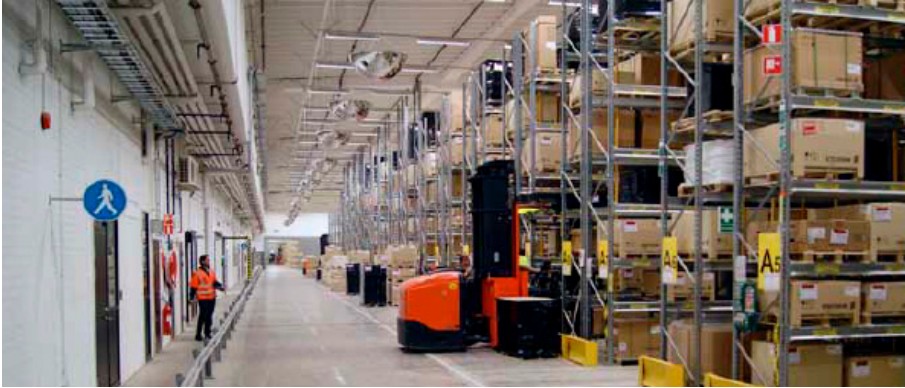

**Figure A3.** A scene of PSS provided for the ICT product provider (Customer I) [67].

**Table A1.** List of all the functions and RSPs in the PSS family.

| # | RSP | Function Name | ICT Product Provider (Customer I) | Construction Company (Customer C) | Whisky Producer (Customer W) | Cost Type | Total Offering Cost |
|---|-----|---------------|-----------------------------------|-----------------------------------|------------------------------|-----------|---------------------|
| | | *Provide access to service center (abstract)* | – | – | – | – | – |
| | | + Take support calls | ✓ (0.22) | ✓ (0.12) | | Proportional | 10 |
| | | + Set up service center access | ✓ (0.22) | ✓ (0.12) | | Fixed | 150 |
| 1 | Availability | *Provide service technician (abstract)* | – | – | – | – | – |
| | | + Conduct preventive maintenance | ✓ (0.22) | ✓ (0.12) | | Proportional | 124 |
| | | + Conduct full vehicle services | ✓ (0.22) | ✓ (0.12) | | Proportional | 150 |
| | | *Provide training (abstract)* | – | – | – | – | – |
| | | + Provide product knowledge | – | – | – | – | – |
| | | ++ Training via e-learning | ✓ (0.13) | | | Fixed | 15 |
| 2 | Productivity | + Training of how to drive | – | – | – | – | – |
| | | ++ Train truck drivers | ✓ (0.13) | ✓ (0.2) | | Semi-fixed | 60 |
| | | ++ Train instructors | ✓ (0.13) | ✓ (0.2) | | Semi-fixed | 520 |
| | | *Provide productivity consulting services (abstract)* | – | – | – | – | – |
| | | + Conduct internal logistics analysis | ✓ (0.4) | ✓ (0.4) | | Proportional | 414 |
| | | *Lease truck (abstract)* | – | – | – | – | – |
| 3 | Service flexibility | + Lease used trucks | ✓ (0.1) | ✓ (0.22) | | Proportional | 100 |
| | | + Lease new trucks | ✓ (0.1) | ✓ (0.22) | | Proportional | 300 |
| | | Provide payback rental plan | ✓ (0.2) | ✓ (0.45) | | Proportional | 600 |
| | | *Provide a counterbalance truck (abstract)* | – | – | – | – | – |
| | | *+ Provide a powerful truck (abstract)* | – | – | – | – | – |
| | | ++ Provide large electric truck | ✓ (0.18) | | ✓ (0.2) | Proportional | 600 |
| 4 | Product flexibility | ++ Provide a certain type diesel truck | ✓ (0.12) | | | Proportional | 400 |
| | | *+ Provide a compact truck (abstract)* | – | – | – | – | – |
| | | ++ Provide small electric truck | ✓ (0.3) | | | Proportional | 200 |
| | | *Provide safety consulting services (abstract)* | – | – | – | – | – |
| | | + Conduct risk analysis of incident | ✓ (0.17) | ✓ (0.15) | ✓ (0.2) | Proportional | 300 |
| | | + Conduct risk analysis of incident with a proposal for improvement | ✓ (0.17) | | | Proportional | 23 |
| | | + Conduct safety inspection | ✓ (0.17) | ✓ (0.15) | ✓ (0.2) | Proportional | 552 |
| 5 | Safety | *Provide IT services (abstract)* | – | – | – | – | – |
| | | + Set up infrastructure of IT services | ✓ (0.17) | ✓ (0.1) | ✓ (0.13) | Fixed | 83 |
| | | + Regulate access to trucks | ✓ (0.17) | ✓ (0.1) | ✓ (0.13) | Proportional | 15 |
| | | + Provide real-time truck positioning data | ✓ (0.17) | ✓ (0.1) | ✓ (0.13) | Proportional | 60 |

Note: A "+" symbol represents the upper functions, while "++" represents the lower functions in the function hierarchy. Each "✓" means the provision of the function to the particular customer.

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
