# Peer review of "Using Product/Service-System Family Design for Efficient Customization with Lean Principles: Model, Method, and Tool"

_sustainability, doi:10.3390/su12145779_

Round 1
Reviewer 1 Report
Dear Authors,
You have done a good job, well done.
Please find below my comments which I believe will enhance your paper presentation:
- It is not clear what problem are you trying to solve in industry. It will be good to clarify that at the end of the introduction section. A list of highlights to answer the following:
- Why this approach is important to the industry?
- What will be the added value if a company consider this model?
- In line 99, please type the equation as it is not clear.
- Please check the third line in table 1 where there are two terms that are not clear (poor resolution)
- I suggest formatting table 3 in a portrait format rather than this landscape format and then move it to the appendix of the paper.
- There is no consistency in figures presentation (for example, check figures 1, 2 and 6)
Regards
Author Response
Please see the separate file.

Reviewer 2 Report
This is an interesting paper that looks at how the design of a PSS family can be supported by applying lean principles. To reach this purpose, the authors propose an object model and a methodology (composed of methods and tools such as service CAD). The methodology is demonstrated through an industrial case study in collaboration with a logistic service provider.
The paper tackles a concrete problem (both in research and industry) and is well structured and written in English.
Before final publication, I think the authors should make an effort to provide more clarity about:
- PROBLEM STATEMENT: there are some logical connections in the introduction I would like the authors to be more precise about:
- reading line 42, it seems to me that the main problem this paper focuses on is on the customization of PSS.
- But then reading line 55, I read “Instead of PSS customization, PSS family design will be performed since it aims to improve the entire set of offerings holistically. Also, the method will build upon lean principles”
- So, I have been a little bit confused whether designing a PSS family is an alternative to customization, or if it is an alternative means to achieve customization (almost like in product platform development, where mass customization is a way of achieving customization efficiently). In my opinion, it is more the latter, but I would like the authors to elaborate on this.
- If this is the case, the authors should make clearer the connection between customization and PSS family design, also highlighting which is the design strategy to achieve customization that they are challenging (in other words, which is the design strategy against which the design of a PSS family is claimed to be superior for)
- After the authors have provided these clarifications, I would suggest the authors to revise the title in order to highlight the core contribution of the paper. I felt the title a little bit general.
- RELATED WORK: since the paper focuses on the design of PSS families (reading the title), I was surprised of not finding much literature regarding the design of product families. The authors provide this sentence, which would deserve some extension: “For product customization, the concept of mass 79 customization (MC) [24,25] is already prevalent throughout industry. More specifically, for instance, 80 a model named the “product family architecture” and its associated methods have been developed”. I think the authors will find good references in the works of Simpson and Roger Jiao (Jiao, J. R., Simpson, T. W., & Siddique, Z. (2007). Product family design and platform-based product development: a state-of-the-art review. Journal of intelligent Manufacturing, 18(1), 5-29). In particular, I would like the authors to focus on the following aspects:
- The benefit of adopting a product family approach to efficiently design for mass customization
- The challenges related to a product family approach, for example the risk of sub-optimization of each single product variant in order to achieve efficiency at product platform level.
- The similarities and differences between designing for a product family and a PSS family, taking into account benefits and challenges.
- RESEARCH METHOD: I have read section 3 in the paper as a little general, while looking at the appendixes it seems to me that the paper is based on empirical studies conducted in industrially-driven cases (e.g. the logistic service provider reported in section 5.1.). I felt the authors are missing an opportunity to provide a fuller account on these empirical studies (e.g. number of cases, number and role of the industrial practitioners involved in the studies). I suggest the authors to provide a fuller account of the details of the empirical studies in this section (perhaps using a table). This would strengthen the validity of the research followed, as well as its results.
- MODEL, METHOD and TOOL: while I found this section well written, I am missing how the proposed model focuses more on the design of a “family” rather than a single individual PSS. For example, I have read the information object model (figure 3), and the steps provided in section 3.3.2. as a well described outline of a PSS methodology. However, I felt that the focus on the design of a “family” (representing the novelty of this paper) was slowly disappearing in this section. I suggest the authors to provide more clarity about how the proposed approach introduces novel elements compared to the design of a “one-off” PSS design, which take into account the design of a family. I think the authors can find inspiration from the papers from Simpson and Roger Jiao I suggested before.
- DISCUSSION: I found this section well written. However, I would like the authors to elaborate more on one possible risk of applying lean principles in the design of PSS: s the authors correctly write, lean is focusing more on removing muda and mura (improving efficiency), while the design of PSS implies also a stronger focused on value creation, which implies an explorative and iterative activity together with customers (combining a red ocean and blue ocean strategy, if the authors are familiar with the book). This may compromise efficiency and risk in the process. I would like the authors to elaborate on this apparent contradiction, and about what mitigation strategies can be applied to assess whether a design work can be considered “waste” (e.g. rework) or a value adding activity (e.g. iteration). The authors can take inspiration from literature regarding value modelling and lean product development, such as: Bertoni, A., Bertoni, M., Panarotto, M., Johansson, C., & Larsson, T. C. (2016). Value-driven product service systems development: Methods and industrial applications. CIRP Journal of Manufacturing Science and Technology, 15, 42-55.

Author Response
Please see the separate file.

Round 2
Reviewer 1 Report
Dear Authors
Thank you addressing my comments.
I am satisfied with your replies.
Regards